# COVID-19 Vaccination and Related Determinants of Hesitancy among Pregnant Women: A Systematic Review and Meta-Analysis

**DOI:** 10.3390/vaccines10122055

**Published:** 2022-11-30

**Authors:** Antigoni Sarantaki, Vasiliki Evangelia Kalogeropoulou, Chrysoula Taskou, Christina Nanou, Aikaterini Lykeridou

**Affiliations:** 1Midwifery Department, University of West Attica, 12243 Athens, Greece; 2Free-Lancer Midwife, 15123 Athens, Greece; 3Alexandra Maternity Hospital, 11528 Athens, Greece

**Keywords:** COVID-19, vaccination, hesitancy determinants, pregnant women

## Abstract

Mass vaccination against COVID-19 is necessary to control the pandemic. COVID-19 vaccines are now recommended during pregnancy to prevent the disease. A systematic review of the literature in the electronic databases PubMed and EMBASE was performed and we aimed to investigate the attitude of documents towards COVID-19 vaccination and the prognostic factors of vaccination hesitation. A meta-analysis was also conducted to estimate the overall percentage of pregnant women who were willing to be vaccinated or had been vaccinated against COVID-19. A total of 18 studies were included in the review and meta-analysis. The acceptance rate of vaccination against COVID-19 among pregnant women ranged from 17.6% to 84.5%. The pooled proportion of acceptance of vaccination against COVID-19 in pregnant women was 0.53 (95% CI: 0.44–0.61). Predictors of acceptance of COVID-19 vaccination were older age, White race, occupational status, higher level of education, comorbidities, third trimester of pregnancy, influenza vaccination, knowledge about COVID-19, and confidence that vaccines for COVID-19 are safe and effective. The prevalence of COVID-19 vaccination in pregnant women is low. Targeted information campaigns are needed to increase vaccine education in this population.

## 1. Introduction

Pregnancy is considered a state of relative immunosuppression with a decrease in cellular immunity and possible susceptibility to infections. Also, there are changes in hormone levels, such as human chorionic gonadotropin (hCG), progesterone, and cortisol. In addition, the increase in uterus size causes the diaphragm to increase by 4 cm, widening the transverse diameter of the chest by 2 cm and affecting lung capacity. Furthermore, the immaturity of the immune system of fetuses and newborns makes them more vulnerable to infections. Therefore, pregnant women and neonates could be considered high-risk groups for infection during the current pandemic [1].

Although COVID-19 infection increases the risk of severe morbidity and mortality in pregnancy, pregnant women were not included in the initial vaccine trials which aimed to explore the efficacy and safety of vaccines against this novel virus. This has resulted in a lack of data on the safety of vaccination in pregnant women. The Centers for Disease Control and Prevention (CDC) collected reports through the National Notifiable Diseases Surveillance System (NNDSS) between 22 January and 3 October 2020, and a total of 1,300,938 women of reproductive age tested positive for SARS-CoV-2 [2]. Data on pregnancy status were available for 461,825 (35.5%) women, of whom 6.6% were pregnant. Among 461,825 women of reproductive age (15–44 years), 409,462 (88.7%) were symptomatic. Among all symptomatic women, 23,434 (5.7%) were pregnant. After adjustment for age, ethnicity/nationality, and underlying medical conditions, pregnant women were significantly more likely to be admitted to an Intensive Care Unit (ICU) [adjusted risk ratio (aRR) = 3.0; 95% confidence interval (CI) = 2.6–3.4], to need invasive ventilation (aRR = 2.9; 95% CI = 2.2–3.8), to receive extracorporeal membrane oxygenation (ECMO) (aRR = 2.4; 95% CI = 1.5–4.0) and to die (aRR = 1.7; 95% CI = 1.2–2.4), compared to non-pregnant women [2].

Unvaccinated pregnant women with a symptomatic COVID-19 infection are at high risk for preterm delivery, admission to an Intensive Care Unit (ICU), and invasive ventilation [3]. Also, unvaccinated women with symptomatic COVID-19 infection during pregnancy are at higher risk of death than non-pregnant women with a symptomatic COVID-19 infection [2]. In addition, unvaccinated pregnant women have a higher risk of hospitalization for COVID-19 than vaccinated pregnant women [4]. Anti-SARS-CoV-2 immunoglobulins confer immunity to neonates, and vaccines do not cause adverse effects associated with them [5,6].

Engjom et al. [7] studied the characteristics of 214 pregnant women with COVID-19 in five Nordic countries from 1 March to 30 June 2020. Out of a total of 214, 56 had to be hospitalized due to COVID-19 disease. Artymuk et al. [8] assessed the incidence of COVID-19 in pregnant women in the Far Eastern Federal District and the Siberian Federal District over 10 months. During the first year of the SARS-CoV-2 pandemic, 8485 cases of COVID-19 were reported among pregnant women in the Far Eastern Federal District and the Siberian Federal District, representing 5.9% of registered pregnant women and 1.71% of the total affected population. The morbidity rate in pregnant women was 3.02 times higher than in the general population: 5933.2 versus 1960.8 per 100,000 of the population. Pregnant women seem to be associated with greater susceptibility to transmission, presenting more severe forms of the disease or high risk for pregnancy complications [9]. COVID-19 is a risk factor for increased maternal and perinatal morbidity, possibly due to higher preterm birth rates in infected mothers [1].

Thus, several organizations worldwide now recommend vaccination against COVID-19 in pregnancy and women trying to get pregnant, or who may become pregnant in the future, to prevent severe maternal morbidity and adverse birth outcomes. The safety and efficacy of COVID-19 vaccines during pregnancy is a particular concern affecting the decision to vaccinate this vulnerable group [10]. Globally, significant reluctance was observed among pregnant women to accept vaccination against COVID-19. This is likely due to the limited evidence on the safety of vaccines in pregnancy at the start of the pandemic and the conflicting and changing advice given to pregnant women as the pandemic progresses [11]. Vaccination against COVID-19 appears to be highly effective in pregnancy for preventing COVID-19, without increasing the risk of adverse pregnancy outcomes. The risks of COVID-19 outweigh the rare risks of vaccination in pregnancy, and pregnant women should be encouraged to continue vaccination, even in the first trimester. However, the lack of high-quality (i.e., low risk of bias) studies uniformly reporting clinically relevant outcomes and other vaccine types used in low- to middle-income countries is problematic [10]. Regarding the vaccination strategy against COVID-19, the World Health Organization (WHO) recommends vaccination for pregnant women only when the benefits of vaccination outweigh the potential risks, but current recommendations for pregnant women to receive vaccinations against COVID-19 differ from country to country [12].

Therefore, this review aimed to examine pregnant women’s attitudes towards vaccination against COVID-19 and to study predictors of vaccination hesitancy.

## 2. Materials and Methods

This systematic review was conducted in accordance with the Preferred Reporting Items for Systematic Reviews and Meta-Analyses (PRISMA) (Figure 1: Flow Chart).

To search for studies related to the purpose of the review, the method followed was based on the PICOST procedure, where P represents population, I intervention, C comparison, O outcome, S the type of studies (Study Design), and T the year of publication (Timeframe). In more detail, the selection/exclusion criteria of the publications for inclusion in this review were as follows:Population: Studies involved pregnant women or women who had recently given birth.Intervention: Investigating pregnant women’s attitudes towards vaccination against COVID-19 through a questionnaire. Studies investigating pregnant women’s attitudes towards vaccines for diseases other than COVID-19 were rejected.Comparison: Studies compared demographic, social, and occupational characteristics of pregnant women with their attitudes toward vaccination against COVID-19.Outcome: The studied outcomes of the publications/studies were the rate of acceptance of the vaccine against COVID-19 and the factors related to hesitancy or refusal of vaccination.Study Design: The studies were primary quantitative studies. Secondary studies (reviews, meta-analyses), case studies, and qualitative studies were not included in the review.Timeframe: The studies must have been published by 31 December 2021 and written in English or Greek. Studies published in a language other than English and Greek were not included. Furthermore, studies in which the full text could not be located were also excluded.

A systematic review of the international literature was performed in the electronic databases PubMed/MEDLINE and EMBASE during the period of July–September 2022. The keywords used to search for studies were: (pregnancy) AND (COVID-19) AND (vaccination) AND (attitude OR hesitancy OR decline OR acceptance OR concern).

A literature search was conducted in the MEDLINE (PubMed) and EMBASE databases following the predefined search strategy (Figure 1). The electronic search yielded 178 abstracts from PubMed and 146 abstracts from EMBASE. Both sets of records were downloaded from each database to the bibliographic software package EndNote X7 (Clarivate Analytics, Philadelphia, PA, USA) and merged into one core database to remove duplicate records and facilitate retrieval of relevant articles. All potentially relevant reports identified after searching other nonelectronic sources were entered into EndNote manually. After the elimination of duplicates, 157 studies were identified. Additionally, seventy-three new studies were removed for missing data, being off topic, and unknown languages (Figure 1).

The titles and abstracts of all identified studies were examined by two reviewers independently (A.S. and V.K.) according to predefined inclusion and exclusion criteria. Review authors were not blinded to the names of the authors, institutions, journal of publication, or results of the studies. All records identified by the searches were primarily checked on the basis of the title and abstract. Records that were obviously irrelevant were excluded, and the full text of all remaining records was obtained. If the relevant information for meeting the inclusion criteria was not available from the abstract and/or title, we obtained the full text of the report. In this way, 32 studies were selected for full-text reading and were assessed independently by the same two reviewers. Articles that did not meet all inclusion criteria after the full-text assessment (N =  9) were excluded from further examination. Figure 1 depicts and summarizes the complete study selection process.

The publications resulting from the literature review were screened as to their title, and those whose title was not compatible with the purpose of the systematic review were rejected. The abstracts of the remaining studies were then read, and those that did not meet the criteria for inclusion in the review were discarded. The remaining studies were searched as full texts and those that did not provide the necessary information regarding the topic and purpose of the review were rejected.

The data selected to be extracted are the following:General characteristics of the study (name of 1st author, year of publication, country of conduct, type of study)Sample characteristics (sample size, average age of pregnant women, gestational week)Outcome (vaccination acceptance/refusal rate)Significant findings (finding, or non-statistically significant association of vaccination acceptance/hesitancy with demographic, social, and occupational characteristics)

### Statistical Analysis

The effect size used was the acceptance rate of vaccination against COVID-19 (Proportion) in pregnant women. The meta-analysis was performed using the metaprop command and the results are presented in the form of a table and plot of findings of the individual studies and overall analysis (forest plot). Study heterogeneity was checked with the I2 test for all studies that provided results. When this was statistically significant (>50%), a random effect model was applied using the DerSimonian and Laird (D+L) method, while a fixed effect model was used otherwise. Publication bias refers to the non-representative publication of research reports, which is not due to the quality of the respective study. The present study did not check for publication bias, either by drawing a funnel plot or with the statistical criterion of Egger’s test, as according to studies on proportional data these tests are not sufficiently adjusted. The assumption that positive results are published more often does not necessarily apply to proportional studies, as there is no clear definition nor consensus on what constitutes a positive result in a proportional meta-analysis. Therefore, it is not recommended in these proportional meta-analyses to conduct publication bias checks [13]. Furthermore, no meta-analysis was performed on factors influencing the vaccination acceptance rate among pregnant women, as the data were highly heterogeneous. The analysis was performed with the statistical package STATA 13. STATA 13 is a statistical software package created in 1985 by StataCorp for data analysis, data management, and graphics. It is a general-purpose statistical software package used in research, especially in the fields of biomedicine and epidemiology.

## 3. Results

From the search through international literature, 324 studies were found. Out of these, 292 were rejected after reading the title, nine after reading the abstract, and five after reading the full text. Ultimately, 18 studies were included in the present review (Figure 1: Flow Chart).

In Table 1 the characteristics of the included studies are demonstrated (author, year, research type, methodology, and main findings). Table 2 shows the acceptance rate of vaccination against COVID-19 among pregnant women according to the studies included in the review and related factors of hesitancy (i.e., maternal age, gestational age, educational level, marital status, race/ethnicity, and presence of comorbidities). As shown in the table, the acceptance rate of vaccination against COVID-19 among pregnant women ranged from 17.6% [14,15] to 84.5% [15].

Of the six studies that investigated whether age is a predictor of COVID-19 vaccination acceptance among pregnant women, only two found age to be a predictor [15,18]. More specifically, in the study by Ghamri et al. [18], it was found that for every year of increase in the age of a pregnant woman, the probability of accepting vaccination increases by 2% [OR: 1.02; 95% CI: 1.018–1.036). Moreover, in Levy’s et al. study, pregnant women aged 18–24 years were found to be 0.35 times less likely to be vaccinated against COVID-19 than women aged 31–35 years [20].

Out of the nine studies that investigated whether education level is a predictor of acceptance of COVID-19 vaccination among pregnant women, seven studies showed that education level is a predictor of acceptance [16,18,20,24,28,30,31]. More specifically, in the study by Battarbee et al. [16], it was found that pregnant women who had a graduate school degree were 2.4 times more likely (OR: 2.4; 95% CI: 1.3–4.7) to accept being vaccinated against COVID-19 compared to pregnant women who were high school graduates. In a study by Ghamri et al. [18], pregnant women with primary education were 2.8 times more likely (OR: 2.853; 95% CI: 1.207–6.745) to accept vaccination against COVID-19 than illiterate pregnant women. In a study by Levy et al. [20], female high school graduates were 0.14 times less likely (OR: 0.14; 95% CI: 0.07–0.25) to accept being vaccinated against COVID-19 in relation to pregnant women who had a Bachelor’s degree. In a study by Riad et al. [24], pregnant women with Master’s degree and a Ph.D. were 5.99 times more likely (OR: 5.99; 95% CI: 1.12–32.16) to accept getting a COVID-19 vaccine dose in relation to pregnant women who had received only basic education. Also, in a study by Skjefte et al. [28], in comparison with pregnant women who were high school graduates, pregnant women with lower than secondary education were 0.76 times less likely to be vaccinated (OR: 0.76; 95% CI: 0.58–0.99), while pregnant women who held university degrees (OR: 1.25; 95% CI: 1.03–1.53) and had completed postgraduate studies (either MSc or Ph.D. (OR: 1.26; 95% CI: 1.10–1.44) were 1.25 and 1.26 times more likely to get a COVID 19 vaccine, respectively.

Only the study by Riad et al. [24] showed that pregnant women who had a partner were 5.43 times more likely to accept being vaccinated against COVID-19 than those who were single.

Six out of the ten studies showed that occupational status is a predictor of acceptance of COVID-19 vaccination among pregnant women [16,18,20,26,28,30]. Employed pregnant women were found to be more likely to accept being vaccinated against COVID-19 than pregnant women who were unemployed/in charge of housekeeping [16,18,26,32]. Also, full-time employed women were more likely to get vaccinated against COVID-19 than part-time employed pregnant women [20,30]. Finally, pregnant physicians [28] and health professionals [26] were more likely to be vaccinated against COVID-19.

Three out of the five studies showed that ethnicity is a predictor of COVID-19 vaccination acceptance among pregnant individuals [16,20,29]. Relative to pregnant white women, those who were black [16,20,29] and Hispanic [16,20,29] were less likely to be vaccinated against COVID-19.

Only two [18,28] out of seven studies showed that the presence of underlying diseases in pregnant women increases with statistical significance the likelihood of accepting vaccination against COVID-19 among this population.

Two studies found that pregnant women in their third trimester were more likely to accept vaccination against COVID-19 than pregnant women in early pregnancy [24,31].

Table 3 shows the results of the studies included in the review regarding influenza vaccination as a factor of COVID-19 vaccination acceptance in pregnancy. Six studies [16,18,20,28,29,30] stated that pregnant women who had been vaccinated against influenza either during their pregnancy or in the previous year were more likely to accept the vaccination against COVID-19 than pregnant women who had never received an influenza vaccination.

Table 4 shows the results of the studies included in the review regarding the existence of a positive COVID-19 test as a factor of vaccination acceptance against COVID-19 among pregnant women, their knowledge regarding COVID-19, and their confidence that vaccines are safe and effective. One study found that pregnant women who had tested positive for COVID-19 were more likely to be vaccinated against COVID-19 than those who never had a positive COVID-19 test [18].

Likewise, two studies found that a higher level of knowledge about COVID-19 or the vaccine was a predictor of acceptance of vaccination against COVID-19 among pregnant women [29,30]. Finally, the perception that the COVID-19 vaccine is safe [23,28] and effective [16,28] was found to be a predictor of vaccination acceptance against COVID-19 in the participating pregnant population.

### Quantitative Synthesis of Studies–Meta-Analysis

The 18 studies included in the review were used to investigate the pooled acceptance proportion of vaccination against COVID-19 in pregnant women. Since a statistically significant heterogeneity among the studies was detected (I^2^ = 99.29%, *p* < 0.001), an analysis was performed using the random effect model. This analysis revealed that the pooled proportion of vaccination uptake against COVID-19 in pregnant women was 0.53 (95% CI: 0.44–0.61). (Figure 2).

## 4. Discussion

The present systematic review and meta-analysis included 18 studies that met certain inclusion and exclusion criteria and aimed to estimate the acceptance of COVID-19 vaccination among pregnant women. Additionally, we intended to investigate factors associated with vaccination acceptance, as well as reasons for vaccination hesitancy.

The acceptance rate of vaccination against COVID-19 among pregnant women ranged from 17.6% to 84.5%. The pooled proportion of vaccination uptake against COVID-19 in the pregnant population was 0.53 (95% CI: 0.44–0.61). The results of our study are similar to the results of two earlier meta-analyses [33,34] and found that the global prevalence of pregnant women receiving the vaccine for COVID-19 was approximately 49–54%. However, in a meta-analysis investigating the pooled proportion of women who had been vaccinated against COVID-19, it was 29% [35].

Acceptance of COVID-19 vaccination was associated with several factors, including older age, ethnicity, occupational status, higher level of education, presence of comorbidities, advanced gestation, influenza vaccination, knowledge about COVID-19, and confidence that COVID-19 vaccines are safe and effective. More specifically, older age was found to be associated with higher acceptance of COVID vaccines [18,20]. This finding is plausible because pregnancy at advanced maternal age is known to be a risk factor for adverse outcomes, such as higher rates of NICU admission for the neonate, preterm birth, miscarriage, preeclampsia, low birth weight, worse Apgar scores, and cesarean deliveries. In addition, older age is associated with higher mortality due to COVID-19. It is probable that older pregnant women face COVID-19 with added fear, resulting in their higher acceptance of COVID-19 vaccination [33,34,35].

Moreover, in our review, the rate of COVID-19 vaccination acceptance was highest among White and Asian pregnant women and lowest among those of Black and Hispanic origin [16,20,29]. Latins and Black or African American race are associated with refusal of vaccination against COVID-19 during pregnancy [16,20,29]. Similar racial and ethnic disparities have been reported for the uptake of other recommended vaccinations during pregnancy, such as tetanus, influenza, and pertussis, with black and Hispanic women having the lowest vaccination coverage [33,34,35].

The safety and effectiveness of COVID-19 vaccines during pregnancy is a particular concern affecting vaccination uptake by this vulnerable population [10]. Regarding the vaccination strategy against COVID-19, the World Health Organization (WHO) recommends vaccination for pregnant women only when the benefits of vaccination outweigh the potential risks, but current recommendations for pregnant women vaccinations against COVID-19 differ from country to country. For example, the USA, the UK, and other European countries encourage pregnant women to get vaccinated against COVID-19, but China does not [36]. For ethical and other reasons, evidence from randomized controlled trials (RCTs) on the safety and efficacy of COVID-19 vaccines for the pregnant population is scarce. Therefore, the real-world study (RWS) on the safety and efficacy of COVID-19 vaccines for pregnant women may provide additional evidence [36].

In the recent systematic review and meta-analysis by Prasad et al. [10], which included 23 studies with 117,552 pregnant women vaccinated against COVID-19, it was found that the efficacy of mRNA vaccination against RT-PCR confirmed SARS-CoV-2 seven days after the second dose was 89.5% (95% CI = 69.0–96.4%, 18,828 vaccinated pregnant women). The risk of stillbirth was significantly lower in vaccinated pregnant women by 15% (pooled OR = 0.85; 95% CI = 0.73–0.99; 66,067 vaccinated vs. 424,624 unvaccinated). There was no evidence of a higher risk of adverse outcomes, including miscarriage, preterm birth, placental abruption, pulmonary embolism, postpartum hemorrhage, maternal death, ICU admission, lower birth weight, or NICU admission (*p* >  0.05 for all). Therefore, vaccination with COVID-19 mRNA in pregnancy appears to be safe and associated with a reduction in stillbirth.

Ma et al. [36] explored the safety and efficacy of vaccines against COVID-19 in pregnant women through the RWS systematic review and meta-analysis. The researchers included a total of six studies. An analysis of the studies found that vaccination prevented the pregnant population from SARS-CoV-2 infection (OR = 0.50; 95% CI = 0.35–0.79) and hospitalization related to COVID-19 (OR = 0.50; 95% CI = 0.31–0.82). mRNA vaccines could reduce the risk of infection in pregnant women (OR = 0.13; 95% CI = 0.03–0.57). Also, no adverse effects of vaccination against COVID-19 were found in pregnant, fetal, or neonatal outcomes.

A question of particular interest is whether the risk of miscarriage increases after vaccination against COVID-19 in early pregnancy. This is particularly important, as up to 40% of pregnancies are unintended and may remain unrecognized until the 4th–8th week of gestation, and thus inadvertent vaccination in early pregnancy is likely to be common [37]. The mRNA vaccine elicits both antibody and cellular immune responses. Given the importance of T-cell suppression in placental development and fetal well-being, concern has been raised that the vaccine may increase the risk of miscarriage [38]. Social media is full of reports that have fueled this concern, and many pregnant women have cited this fear as their main reason for hesitating to vaccinate. Data from the meta-analysis by Prasad et al. [10] do not support such concerns.

Also, vaccination against COVID-19 is associated with a lower incidence of stillbirth. COVID-19 in pregnancy is associated with an increased risk of stillbirth, particularly during the period of Delta variant dominance [10]. A population-based study in Scotland found that after SARS-CoV-2 infection in unvaccinated pregnant women, the perinatal mortality rate was 22.6 per 1000 births, whereas, in contrast, no vaccinated pregnant women with primary infection suffered a perinatal death [39].

In Brazil, there has been a report of maternal death after vaccination with Astra Zeneca, and other cases are also reported in the published literature. With the Oxford/Astra Zeneca virus vaccine, there is a rare risk of vaccine-induced immune thrombotic thrombocytopenia [40]. This has prompted some countries (UK, Canada, and USA) to exclude this vaccine in people under 40 years of age. These very rare complications should not discourage the scientific community, health workers, and policymakers from disseminating information about the clear benefits of vaccination against COVID-19 in pregnancy to both mother and newborn [10].

Similarly, cases of myocarditis have been reported after vaccination with mRNA, estimated to occur at two cases per million women and 10 cases per million men aged 18–40 years. Such reactions are typically mild and rapidly self-limiting and occur more often in association with COVID-19 infection. However, no cases of myocarditis have been reported after vaccination in pregnant women [10].

Globally, significant reluctance was observed among pregnant women to accept vaccination against COVID-19. This is likely due to the limited evidence on the safety of vaccines in pregnancy at the start of the pandemic and the conflicting and changing advice given to pregnant women as the pandemic progresses [11]. Vaccination against COVID-19 appears to be highly effective in pregnancy for preventing infection, without increasing the risk of adverse pregnancy outcomes. The risks of COVID-19 outweigh the rare risks of vaccination in pregnancy, and pregnant women should be encouraged to continue vaccination, even in their first trimester. However, the lack of high-quality (i.e., low risk of bias) studies uniformly reporting clinically relevant outcomes, as well as reporting other vaccine types used in low- to middle-income countries, is rather challenging.

Confidence in COVID-19 vaccines and fewer concerns about the safety and side effects of COVID-19 vaccines are predictors of acceptance of COVID-19 vaccination [16,28]. Similar factors, such as confidence in the safety and effectiveness of COVID-19 vaccines, confidence in the information received about vaccination against COVID-19, confidence in childhood vaccines, and influenza vaccination in the previous year, are associated with a higher rate of intention of pregnant women to receive a COVID-19 vaccine [16,18,20,28,29,30]. In general, high levels of information and knowledge about COVID-19 vaccines reduce fear and have a significant effect on a pregnant woman’s decision to get vaccinated against COVID-19 [33,34,35]. A recent systematic review and meta-analysis found that vaccination for COVID-19 protects pregnant women from SARS-CoV-2 infection and COVID-19-related hospitalization and has no adverse effects on pregnant women, fetuses, or neonates. Since COVID-19 vaccines have been shown to be safe and effective in pregnant women, policymakers should use this information to improve confidence in COVID-19 vaccines and reduce hesitancy [33,34,35].

The present review and meta-analysis is characterized by some limitations. Initially, publications were searched only in two databases, and studies published only in English and Greek were included. Another limitation of the review is the heterogeneity of the studies in terms of their design and the study population. Finally, the quality of the publications was not thoroughly assessed.

Previous studies have stated COVID-19 vaccine intention among pregnant women; our study is among the first to compare vaccination behavior in the light of vaccine acceptance and hesitancy attitudes toward vaccination in this specific population.

## 5. Conclusions

Pregnant individuals are at increased risk of severe COVID-19 infection. Although vaccination is suggested, COVID-19 vaccination rates are lower among pregnant women compared to the non-pregnant population.

The present analysis shows that approximately one in two pregnant women is willing to be vaccinated against COVID-19. Numerous factors, including advanced maternal age, White race, employment, educational level, underlying diseases, influenza vaccination, knowledge about COVID-19 infection, and confidence that vaccines for COVID-19 are safe and effective, are associated with COVID-19 vaccination acceptance.

COVID-19 vaccine acceptance during pregnancy is not widespread and public health intervention will be vital to continue to increase vaccine coverage. Efforts should be made in order to encourage uptake and facilitate access to vaccination. These actions may contribute to higher uptake in pregnant populations of lower socioeconomic status and all those associated with lower vaccine acceptance.

The safety of the COVID-19 vaccine is a principal concern for pregnant people when deciding to receive the vaccine. As pregnant and lactating women were not included in the COVID anti-viral and vaccine trials, it is imperative that safety information is communicated to this high-risk population as it occurs, in order to advance empirical evidence.

Information delivery methods about the benefits of vaccination during pregnancy should be improved to increase the likelihood that someone will act on it, as the elevated risk of COVID-19 disease in pregnancy is either poorly understood or underrated.

## Figures and Tables

**Figure 1 vaccines-10-02055-f001:**
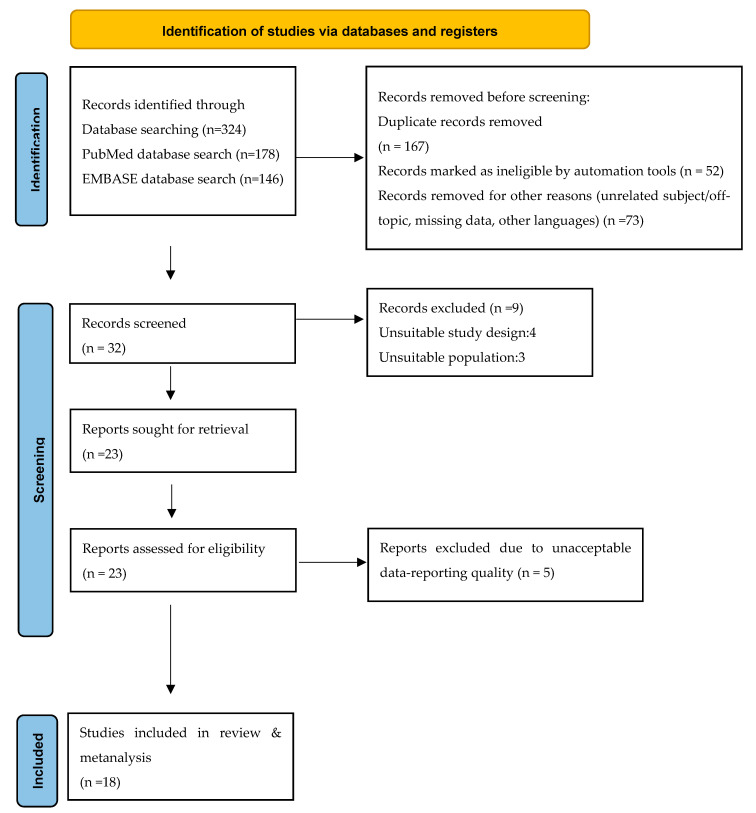
PRISMA flow diagram. Visualization of the process involving identification of records from databases, screening of records, assessing reports for eligibility, inclusion of eligible studies, and exclusion of non-eligible reports. The number of records or reports in each step of the process is shown in brackets.

**Figure 2 vaccines-10-02055-f002:**
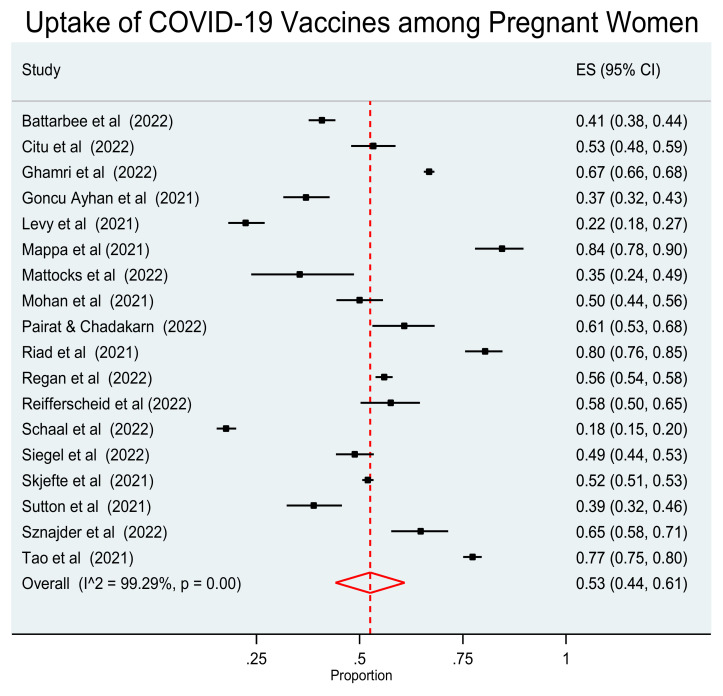
Forest plot of studies reporting vaccination acceptance against COVID-19 in pregnant women. Vertical ticks and horizontal lines show the mean effect and 95% confidences interval for each study. The red diamond at the bottom shows the cumulative effect with 95% confidence intervals.

**Table 1 vaccines-10-02055-t001:** Characteristics of studies included in review & metanalysis.

Author/s (Year, Country)	Research Type	Time Period	Aim	Research Tools	Sample	Results
					Ν/Age/Gestational Age	
Battarbee et al. (2022, USA) [16]	cross-sectional survey	9 August–10 December 2020	To evaluate pregnant women’s attitudes toward COVID-19 illness and vaccination and identify factors associated with vaccine acceptability	Pregnant women filled out questionnaires when enrolled in a prospective COVID-19 cohort	Ν = 915/81% was 18–34 yrs/<28 weeks of gestation	Although most pregnant women worried about COVID-19 illness, <50% were willing to get vaccinated during pregnancy. Racial and ethnic disparities in plans to accept the COVID-19 vaccine highlight the need to prioritize strategies to address perceived barriers among high-risk groups at high risk for COVID-19.
Citu et al. (2022, Romania) [17]	cross-sectional study	1 January–1 May 2022	Το determine the scale of acceptance of the COVID-19 vaccination campaign among pregnant women in Romania, as well as the variables affecting their choices	online survey including standardized and unstandardized questionnaires	Ν = 345/Not specified/not specified	The statistically significant determinant factors for COVID-19 vaccination acceptance were the urban area of residence, having a higher level of education, the third trimester of pregnancy, trusting the government, being a frequent traveler, fearing the severity of COVID-19, the higher availability of COVID-19 vaccines nearby, and seeing more people getting vaccinated.
Ghamri et al. (2022, Saudi Arabia) [18]	cross-sectional study	July–September 2021	To determine the level of acceptance of COVID-19 vaccination and detect the factors that influence vaccine acceptance among pregnant women in Saudi Arabia	web-based study	Ν = 5307/˃18/multi-stage sampling	High acceptance of COVID-19 vaccination. The two major reasons for refusal were concerns about a lack of data on COVID-19 vaccination safety and the possibility of harming the fetus.
**Authors (Year, country)**	**Research type**	**Period**	**Aim**	**Research Tool**	**Sample**	**Results**
					**Ν/Age/Gestational age**	
Ayhan Goncu et al. (2021, Τurkey) [19]	prospective observational study	1 January–1 February 2021	To determine vaccine acceptance and hesitancy attitudes toward coronavirus disease 2019 (COVID-19) vaccines in pregnant women	face to face interviews with 40 questions	Ν = 300/ 27.99 (±5.6) yrs/ 28.74 (±8.88) wks	Low acceptance of COVID-19 vaccination in the sample of pregnant women. Concern about vaccine safety was the major reason for hesitancy.
Levy et al. (2021, USA) [20]	cross-sectional study	14 December 2020–14 January 2021	To describe the COVID-19 vaccine acceptance rate among pregnant women	questionnaires of 31 questions	Ν = 662/82.9% >30 yrs/during the nuchal translucency or anatomic survey sonogram appointment	Young age, Black or African American race, Hispanic ethnicity, having less than a Bachelor’s degree, and declining the seasonal influenza vaccine were associated with nonacceptance of COVID-19 vaccination in pregnancy. Trust in the information received about vaccinations was the strongest predictor of COVID-19 vaccination acceptance.
Mappa et al. (2021, Italy) [15]	prospective observational study	December 2020	To evaluate the propensity of a population of Italian women to receive the vaccine and its psychological impact	a multi-section questionnaire, State–Trait–Anxiety–Inventory (STAI)	Ν = 161/not specified/not specified	The majority of pregnant women were considered to have a positive attitude toward the SARS-CoV-2 vaccine. The vaccine campaign seems to increase the maternal level of anxiety and this increase is more marked by a negative attitude toward the vaccine.
Mattocks et al. (2022, USA) [21]	cross-sectional study	1 January–31 May 2021	To examine pregnant veterans’ acceptance of COVID-19 vaccines, along with perceptions and beliefs regarding vaccine safety and vaccine conspiracy beliefs	telephone surveys (~45 min in length)	Ν = 92/not specified/at 20 wks & 3 months postpartum	Those who received a vaccine had significantly greater vaccine knowledge and less belief in vaccine conspiracy theories. Greater knowledge of vaccines in general and fewer beliefs in vaccine conspiracies were the strongest predictors of acceptance of a COVID-19 vaccine during pregnancy.
**Authors (year, country)**	**Research type**	**Period**	**Aim**	**Research tool**	**Sample**	**Results**
					**Ν/Age/Gestational age**	
Mohan et al. (2021, Qatar) [22]	cross-sectional study	15 October–15 November 2020	To explore attitudes to COVID-19 vaccination among perinatal women	online survey; a composite questionnaire, a validated vaccine hesitancy measurement tool called VAX	Ν = 341/ 69.8% were 26–35 yrs/not specified	The sample exhibited a vaccine hesitancy rate of 25% toward COVID-19 immunization. The group’s main concerns were infection risks, and the main factor determining vaccine hesitancy was vaccine-specific safety concerns. Previous vaccine “acceptors” showed vaccine hesitancy to COVID-19 immunization A third of the group cited the non-availability of the vaccine as a concern.
Pairat & Phaloprakarn (2022, Thailand) [23]	prospective survey	1 July–30 September 2021	To determine the rates and associated factors of accepting attitudes toward COVID-19 vaccination during pregnancy among pregnant Thai women and their spouses, and to evaluate the actual rate of vaccination during pregnancy among these women	Written self-answered questionnaires were completed by eligible couples (pregnant women and their husbands - separately) during their hospital visit.	Pregnant women Ν = 171/median age = 28 (23–33 yrs)/26 wks (18–31 wks of gestation)	Having a husband who favored COVID-19 vaccination for his wife was independently associated with COVID-19 vaccine acceptance among pregnant women.
Riad et al. (2021, Czech Republic) [24]	cross-sectional survey-based study	1 August–31 October 2021	To evaluate the attitudes of pregnant and lactating Czech women (PLW) towards COVID-19 vaccines and the determinants of their attitudes	a self-administered questionnaire (SAQ) consisting of 32 close-ended items, adapted from previous instruments used for the same purpose	Ν = 278 pregnant/ 51.5% was 19–31 yrs/85.6% were in the 3rd trimester	The overall COVID-19 vaccine acceptance (both immediate and delayed) level was substantially high, with a significant difference between pregnant women. The trimester, education level, employment status, and previous live births were significant determinants for COVID-19 vaccine acceptance among the target population.
Regan et al. (2022, USA) [25]	cross-sectional survey	1 December 2020–31 July 2021	To evaluate the acceptance of any dose of the COVID-19 vaccine during pregnancy	online questionnaire	Ν = 2213/not specified/not specified	Participants were more likely to receive or plan to receive the COVID-19 vaccine if they had group prenatal care, were employed in a workplace with a policy recommending vaccination, and believed COVID-19 vaccines are safe.
**Authors (Year, Country)**	**Research Type**	**Period**	**Aim**	**Research tool**	**Sample**	**Results**
					**Ν/Age/Gestational age**	
Reifferscheid et al. (2022, Canada) [26]	cross-sectional study	28 May–7 June 2021	To investigate COVID-19 vaccine uptake and intent among pregnant people in Canada and determine associated factors	online questionnaire	Ν = 193/ 31.0 (±6.2) yrs/ Not specified	Confidence in vaccine safety was the most significant predictor of COVID-19 vaccine acceptance among respondents, and vaccine safety concerns were the most cited reason for not accepting the COVID-19 vaccine during pregnancy.
Schaal et al. (2022, Germany) [14]	cross-sectional study	30 March–19 April 2021	To address COVID-19 vaccination attitudes among pregnant and breastfeeding women including the underlying reasons for their decision	online questionnaire	Pregnant women Ν = 1043/31.8 (±4.3) yrs/ 24.7 (±9.1) wks of gestation	The willingness to be vaccinated was significantly related to the women’s anxiety levels about getting infected and developing disease symptoms. The main reasons for vaccination hesitancy were women’s perception of vaccination-specific information, limited scientific evidence on vaccination safety, and the fear of harming the fetus or infant.
Siegel et al. (2022, USA) [27]	cross-sectional study	1 June–31 August 2021	To characterize attitudes toward novel coronavirus disease 2019 (COVID-19) vaccination and to evaluate factors associated with vaccine uptake among pregnant individuals	a convenience sample of pregnant individuals receiving prenatal care	Ν = 477/ Not specified/ Not specified	Overall, 233 (49.3%) had received or were scheduled to receive a COVID-19 vaccine. Age, White race, non-Hispanic or Latin ethnicity, working from home, and typical receipt of the influenza vaccine were factors associated with COVID-19 vaccination.
Skjefte et al. (2021, 16 countries) [28]	cross-sectional study	28 October–18 November 2020	To assess acceptance of COVID-19 vaccination among pregnant women and mothers of children younger than 18 years old, as well as potential predictors	online questionnaire	Ν = 5282/ 34.4 (±7.3) yrs/ 20.0 (±9.4) wks	Vaccine acceptance was generally highest in India, the Philippines, and all sampled countries in Latin America; it was lowest in Russia, the United States, and Australia. The strongest predictors of vaccine acceptance included confidence in vaccine safety or effectiveness, worrying about COVID-19, belief in the importance of vaccines to their own country, compliance with mask guidelines, trust of public health agencies/health science, as well as attitudes towards routine vaccines.
**Authors (Year, Country)**	**Research Type**	**Period**	**Aim**	**Research tool**	**Sample**	**Results**
					**Ν/Age/Gestational age**	
Sutton et al. (2021, USA) [29]	cross-sectional study	7 January–29 January 2021	To understand vaccine acceptability among pregnant, nonpregnant, and breastfeeding respondents and elucidate factors associated with COVID-19 vaccine acceptance	online questionnaire of 23 items	Pregnant sample: Ν = 216/ 34.0 (±6.0)/ Not specified	Pregnant respondents of non-White or non-Asian races were more likely to decline vaccination than nonpregnant and breastfeeding respondents. Working in healthcare was not associated with vaccine acceptance.
Sznajder et al. (2022, USA) [30]	cross-sectional study	15 May–1 December 2020	Τo examine factors associated with vaccine acceptance	Pregnant women receiving prenatal care completed a questionnaire.	Ν = 196/ 80% aged <35 yrs/ 44% in the 3rd trimester	Women who had received an influenza vaccine within the past year were more likely to be willing to receive the COVID-19 vaccine than women who had never received an influenza vaccine or those who received it more than one year ago. Similarly, women who were employed full-time were more willing to receive the COVID-19 vaccine than women who were not employed full-time.
Tao et al. (2021, China) [31]	cross-sectional study	13–17 November 2020	To explore the acceptance of a COVID-19 vaccine and related factors among pregnant women	questionnaire distributed in 6 hospitals	Ν = 1392/ 55.4% <30 yrs/ 44% in the 3rd trimester	About one quarter of pregnant women have vaccine hesitancy. The acceptance rate was associated with young age, western region, low level of education, late pregnancy, high knowledge score on COVID-19, high level of perceived susceptibility, low level of perceived barriers, high level of perceived benefit, and high level of perceived cues to action.

**Table 2 vaccines-10-02055-t002:** Acceptance rate of vaccination against COVID-19 among pregnant women and related factors of hesitancy.

Authors (Year)	(N) Number of Pregnant Women Who Would Get Vaccinated/Total Sample (%)	Maternal Age as a Factor of COVID-19 Vaccination Acceptance	Gestational Age as a Factor of COVID-19 Vaccination Acceptance	Educational Level as a Factor of COVID-19 Vaccination Acceptance	Race/Ethnicity as a Factor of COVID-19 Vaccination Acceptance	Marital Status as a Factor of COVID-19 Vaccination Acceptance	Occupational Status as a Factor of COVID-19 Vaccination Acceptance	Underlying Diseases/Comorbidities as a Factor of COVID-19 Vaccination Acceptance
		Maternal Age/OR (95% CI)	Gestational Age/OR (95% CI)	Educational Level/OR (95% CI)	Race/Ethnicity /OR (95% CI)	Marital Status/OR (95% CI)	Occupational Status/OR (95% CI)	Underlying Diseases/Comorbidities OR (95% CI)
Battarbee et al. (2022) [16]	374/915 (41%)	18–34 years/0.8 (0.6–1.1) 35–50 years/Reference category		<High school/reference category High school diploma/0.8 (0.4–1.5) Some college or technical school/0.6 (0.3–1.2) College degree/1.5 (0.8–2.8) Graduate school degree/**2.4 (1.3–4.7)**	White/Reference category Black/**0.3 (0.2–0.4)** Hispanic/**0.3 (0.2–0.4)** Other/0.7 (0.3–1.4)		Employed/**1.4 (1.0–1.9)** Unemployed/Reference category	Yes/1.0 (0.7–1.3) No/Reference category
Citu et al. (2022) [17]	184/345 (53.3%)							
Ghamri et al. (2022) [18]	3548/5307 (68%)	≥18 years of age /**1.02 (1.018–1.036)**	All 3 trimesters/ 0.986 (0.981–0.991)	None/reference category Primary school/**2.853 (1.207–6.745)** Secondary school/1.741 (0.804–3.769) University/1.653 (0.772–3.542)			Housewife/Reference category Private sector/**1.328 (1.118–1.577)** Government sector/**1.685 (1.464–19.39)**	Diabetes mellitus/**1.947 (1.344–2.822)** High blood pressure/**2.340 (1.486–3.684)** Heart disease/**2.962 (1.147–7.650)** No/Reference category
Goncu Ayhan et al. (2021) [19]	111/300 (37%)							
Levy et al. (2021) [20]	381/362 (58.3%)	18–24 years/ **0.35 (0.13−0.97)** 25–30 years/ 0.64 (0.40−1.02) 31–35 years/Reference category 36–40 years/ 1.08 (0.74−1.57) >40 years/ 0.74 (0.42−1.30)		High School/**0.14 (0.07−0.25)** College/Reference category MSc/0.86 (0.59−1.26) • Ph.D./1.06 (0.65−1.73)	White/Reference category Black or African American/**0.12 (0.06−0.25)** Hispanic/**0.47 (0.29−0.74)**		Full-time employment/Reference category Part-time employment/**0.44 (0.25−0.7)** Unemployed/**0.58 (0.37−0.92)**	
Mappa et al. (2021) [15]	136/161 (84.5%)							
Mattocks et al. (2022) [21]	22/62 (69%)							
Mohan et al. (2021) [22]	158/316 (50.0%)							
Pairat & Chadakarn (2022) [23]	104/171 (60.8%)			Primary education/Reference category Secondary education/1.12 (0.36–3.47) College/1.65 (0.37–7.37) • Bachelor’s degree or higher/1.19 (0.35–4.04)			Public officer/2.68 (0.79–9.09) Business owner/1.09 (0.37–3.26) Employee/1.17 (0.59–2.33) Unemployed/Reference category	Yes/0.59 (0.29–1.18) No/Reference category
Riad et al. (2021) [24]	254/316 (70.2%)		1st trimester/Reference category 2nd trimester/1.18 (0.19–7.50) 3rd trimester/**6.50 (1.21–35.03)**	Primary education/Reference category Secondary education/3.67 (0.78–17.27) College/2.79 (0.44–17.48) MSc or PhD/**5.99 (1.12–32.16)**		Living with a partner/**5.43 (0.57–52.01)** Without a partner/Reference category	Employed/2.44 (0.66–9.99) Unemployed/Reference category	
Regan et al. (2022) [25]	1238/2213 (55.4%)							
Reifferscheid et al. (2022) [26]	111/193 (55.5%)	≥15 years of age/ 1.01 (0.96–1.06)		≤High school/Reference category Non-university certificate or diploma/0.63 (0.24–1.60) University certificate, Bachelor’s degree, postgraduate degree/1.86 (0.79–4.37)	White/Reference category Minority group/0.83 (0.45, 1.56) Indigenous/3.77 (0.94, 25.20)		Unemployed/Reference category Employed, not high risk/**2.50 (1.10, 5.90)** Employed, other high-risk occupation/**4.03 (1.62, 10.53)** Employed, healthcare worker/**3.48 (1.26, 10.19)**	Yes/0.69 (0.37, 1.31) No/Reference category
Schaal et al. (2022) [14]	184/1043 (17.6%)							
Siegel et al. (2022) [27]	233/477 (49.3%)							
Skjefte et al. (2021) [28]	2747/5282 (52.0%)	18–24 years/0.80 (0.56–1.14) 25–29 years/0.88 (0.63–1.23) 30–34 years/0.93 (0.66–1.30) 35–39 years/0.92 (0.65–1.30) 40–65 years/Reference category		Never attended high school/**0.76 (0.58–0.99)** High school or GED/Reference category College diploma or university degree/**1.25 (1.03–1.53)** Master’s, Professional, or Doctoral degree/**1.26 (1.10–1.44)**		Married/0.92 (0.82–1.03) Unmarried/Reference category	Unemployed/Reference category Medical Doctor/**2.26 (1.54–3.32)** Nurse/0.71 (0.57–0.88) Other healthcare workers/0.78 (0.65–0.94) Non-healthcare workers/**1.13 (1.07–1.23)**	Yes/**1.21 (1.08–1.35)** No/Reference category
Sutton et al. (2021) [29]	86/216 (21.3%)				White/Reference category Black/**0.69 (0.58–0.82)** Hispanic/**0.64 (0.56–0.73)** Asian/0.99 (0.86–1.13)		Healthcare worker/1.03 (0.93–1.13) Non-Healthcare worker/Reference category	Yes/0.96 (0.88–1.06) No/Reference category
Sznajder et al. (2022) [30]	127/196 (65%)	≤35 years/**1.87 (1.20–2.93)** >35 years/Reference category	1st or 2nd trimester/0.75 (0.41–1.37) 3rd trimester/reference category	College degree/**2.79 (1.48, 5.28)** <College degree/Reference category	White/1.05 (0.39, 2.79) Non-white/Reference category Hispanic/2.77 (0.32, 24.19) Non-Hispanic/Reference category		Full-time employment/**2.59 (1.41–4.79)** Part-time employment/Reference category	
Tao et al. (2021) [31]	1077/1392 (77.4%)	≤35 years/**1.87 (1.20–2.93)** >35 years/Reference category	1st trimester/Reference category 2nd trimester/1.39 (0.94–2.04) 3rd trimester/**1.49 (1.03–2.16)**	Less than high school/**2.49 (1.13–5.51)** High school or some college/**2.85 (1.45–5.59)** Bachelor’s degree/1.58 (0.83–3.03) Postgraduate degree/Reference category			Employed/Reference category Housewife/1.06 (0.73–1.54)	Yes/0.66 (0.26–1.69) No/Reference category

**Table 3 vaccines-10-02055-t003:** Influenza vaccination as a factor of COVID-19 vaccination acceptance among pregnant women.

Authors (Year)	Influenza Vaccine	
		OR (95% CI)
Battarbee et al. (2022) [16]	Yes	**2.6 (1.9–3.6)**
	No	Reference category
Ghamri et al. (2022) [18]	Yes	**2.639 (1.847–3.771)**
	No	Reference category
Levy et al. (2021) [20]	Yes	Reference category
	Has not received the vaccine, but plans to receive it	**0.48 (0.28–0.82)**
	No	**0.12 (0.07–0.21)**
Pairat & Chadakarn (2022) [23]	Yes	1.61 (0.75–3.45)
	No	Reference category
Skjefte et al. (2021) [28]	Yes	**3.29 (2.91–3.72)**
	No	Reference category
Sutton et al. (2021) [25]	Yes	**2.25 (1.66–3.05)**
	No	Reference category
Sznajder et al. (2022) [30]	Yes	**5.25 (2.55–10.79)**
	No	Reference category
Tao et al. (2021) [31]	Yes	1.30 (0.74–2.28)
	No	Reference category

**Table 4 vaccines-10-02055-t004:** Positive COVID-19 test, knowledge of COVID-19, and confidence that vaccines are safe and effective as factors in acceptance of vaccination against COVID-19 among pregnant women.

Authors (Year)	Positive COVID-19 Test	Knowledge of COVID-19	Confidence That COVID-19 Vaccine Is Safe	Confidence That COVID-19 Vaccine Is Effective
	OR (95% CI)		OR (95% CI)		OR (95% CI)		OR (95% CI)
Battarbee et al. (2022) [16]							Yes	**11.94 (2.72–52.36)**
		No	Reference category
Pairat & Chadakarn (2022) [23]					Yes	**4.42 (2.19–8.93)**	Yes	**4.96 (2.45–10.05)**
No	Reference category	No	Reference category
Reifferscheid et al. (2022) [26]					Yes	1.81 (0.98, 3.36)		
No	Reference category		
Ghamri et al. (2022) [18]	Yes	**1.721 (1.332–2.223)**						
Νο	Reference category						
Levy et al. (2021) [20]	Yes	0.30 (0.16–0.59)	Yes	1.194 (0.998–1.429)				
Νο	Reference category	No	Reference category				
Skjefte et al. (2021) [28]	Yes	1.04 (0.93–1.17)			Yes	**8.42 (7.44–9.53)**	Yes	**6.68 (5.90–7.26)**
No	Reference category			No	Reference category	No	Reference category
Sutton et al. (2021) [29]	Yes	0.46 (0.21–1.01)						
No	Reference category						
Sznajder et al. (2022) [30]	Yes	0.90 (0.21–3.89)						
No	Reference category						
Mattocks et al. (2022) [21]			Non-stop	**1.76 (1.17–2.64)**				

Riad et al. (2021) [24]			Yes	0.91 (0.34–2.48)				
		No	Reference category				
Tao et al. (2021) [31]			Non-stop	**1.05 (1.01–1.10)**

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
