# Peer review of "COVID-19 Vaccination and Related Determinants of Hesitancy among Pregnant Women: A Systematic Review and Meta-Analysis"

_vaccines, 2022, doi:10.3390/vaccines10122055_

Round 1

Reviewer 1 Report

Comments and Suggestions for Authors

In the present work, Sarantaki et al. try to explain Covid-19 vaccination and related determinants of hesitancy among pregnant women. There are some questions that should be considered.

1. Line 18, please explain ‘CI’.

2. Editing of English language and style is needed. Pease revise the manuscript throughout.

For example, ‘Predictors of acceptance of COVID-19 vaccination were older age, white race, occupational status, higher level of education, comorbidities, 3rd trimester of pregnancy, influenza vaccination, knowledge about COVID-19, and confidence that vaccines for COVID-19 are safe and effective.’

‘beta human chorionic gonadotropin, β-hCG,’

‘The Centers for Disease Control and Prevention (CDC) collected reports through the National Notifiable Diseases Surveillance System (NNDSS) that between 22 January and 3 October 2020, a number of 1,300,938 women of reproductive age tested positive for SARS-CoV-2.’

‘(2Zambrano et al., 2020).’

3. Introduction section is very long, which should be simplified.

4. Lines 87-90, a reference should be added.

5. Line 116, ‘studies in which the full text could not be located were also excluded’, suggesting that some studies were not used for analysis.

6. Lines 117-118, ‘A systematic review of the international literature was performed in the electronic database PubMed,’ suggesting that some studies without in the electronic database PubMed were not used for analysis.

7. Lines 128-133, the serial number should be revised.

8. Lines 153-156, please give the reasons for unconsidered studies.

9. Line 156, is where the Table 1, Lines 246 to 247? Format of this Table should be revised.

10. Lines 241-246, detail information for data analysis software should be provided.

11. Tables 1-13, numbers of references should be provided.

12. Line 385, ‘In conclusion,’ and Line 391, ‘5. Conclusions’ should be incorporated.

13. Format of some references is wrong. For example, 2, 3, 5, 8, 9, 10.

Author Response

Respected Reviewer,

Thank you for your observations.

  1. Line 18, please explain ‘CI’.

Answer: In line 18 “The pooled proportion of acceptance of vaccination against COVID-19 in pregnant women was 0.53 (95% CI: 0.44 – 0.61)”. CI=Confidence interval is a range of values so defined that there is a specified probability that the value of a parameter lies within it.

  1. Editing of English language and style is needed. Pease revise the manuscript throughout.

Answer: Although the writer of the paper is fluent in English as she was born and raised in South Africa, with respect to your comment the manuscript was revised by a professional English translator and is resubmitted with minor linguistic corrections.

3.Introduction section is very long, which should be simplified.

Answer: The introduction counts 792 words while the manuscript without figures and tables sums 4.124 words. This introduction is considered long enough to supply sufficient background information for the reader to understand and evaluate the rationale for the study and its aim.

  1. Lines 87-90, a reference is added.
  2. Line 116, ‘studies in which the full text could not be located were also excluded’, suggesting that some studies were not used for analysis.

Answer: The process of data collection is thoroughly described according to your comment. The studies that were included in the analysis were only the ones that the writers had full access to the content of the manuscript.

  1. Lines 117-118, ‘A systematic review of the international literature was performed in the electronic database PubMed,’ suggesting that some studies without in the electronic database PubMed were not used for analysis.

Answer: The process of data collection is thoroughly revised and defined according to your comment.

  1. Lines 128-133, the serial number should be revised.

Answer: Done

  1. Lines 153-156, please give the reasons for unconsidered studies.

Answer: Done

  1. Line 156, is where the Table 1, Lines 246 to 247? Format of this Table should be revised.

Answer: Done

  1. Lines 241-246, detail information for data analysis software should be provided.

Answer: Done

  1. Tables 1-13, numbers of references should be provided.

Answer: Done

  1. Line 385, ‘In conclusion,’ and Line 391, ‘5. Conclusions’ should be incorporated.

Answer: Done

13. Format of some references is wrong. For example, 2, 3, 5, 8, 9, 10.

Answer: Corrected

We hope the answers we provided to your comments are satisfactory enough.

Thank you for your time and effort in reviewing our paper.

Sincerely,

A.S.

Reviewer 2 Report

I was invited to revise the paper "Covid-19 vaccination and related determinants of hesitancy among pregnant women". It was a systematic review aimed to summarize evidence on attitude towards covid19 vaccination among pregnants and related factors. 

The topic is important for public health and a SR can improve the knowledge on this fields.

Observations:

- The title should me more informative. I suggest "Covid-19 vaccination and related determinants of hesitancy among pregnant women: a systematic review and metanalysis";

- In lines 37-38 Authors stated "Pregnant women were excluded in the initial randomized controlled clinical trials, which aimed to explore the efficacy and safety of vaccines against COVID-19". No trials include pregnant women, except for pregnancy treatment. So Authors should modify this sentence;

- Authors should specify which database were exlplored;

- Too many tables were included. Authors should present a table 1 with study characteristics of all included studies. In addition a secon table can be added regarding vaccination acceptance and related factors. In addition, information about country shoud be added in table 1;

- Statistical analysis should be reported in a separate section of methods;

- Forest-plot (presented without a title!) and quantitative analysis results  should be presented after the qualitative analysis;

- Flowchart of the study should be reported prior results. In addition, Authors should report in figure how many studies were included in the systematic review and how many studies were included in the metanalysis;

- Discussion are poor. Authors did not compare included study and did not compare these results with similar study against different kind of vaccines. 

Author Response

Respected Reviewer,

Thank you for your observations.

- The title should be more informative. I suggest "Covid-19 vaccination and related determinants of hesitancy among pregnant women: a systematic review and metanalysis";

Answer: the title was changed according to your comment

- In lines 37-38 Authors stated "Pregnant women were excluded in the initial randomized controlled clinical trials, which aimed to explore the efficacy and safety of vaccines against COVID-19". No trials include pregnant women, except for pregnancy treatment. So Authors should modify this sentence;

Answer: the sentence was modified

- Authors should specify which database were explored;

Answer: PubMed/MEDLINE and EMBASE were the databases searched and referred to the manuscript

- Too many tables were included. Authors should present a table 1 with study characteristics of all included studies. In addition, a secon table can be added regarding vaccination acceptance and related factors. In addition, information about country shoud be added in table 1;

Answer: The tables were modified according to the comments

- Statistical analysis should be reported in a separate section of methods;

Answer: Done

- Forest-plot (presented without a title!) and quantitative analysis results  should be presented after the qualitative analysis;

Answer: Done

- Flowchart of the study should be reported prior results. In addition, Authors should report in figure how many studies were included in the systematic review and how many studies were included in the metanalysis;

Answer: Done

- Discussion are poor. Authors did not compare included study and did not compare these results with similar study against different kind of vaccines. 

Answer: Discussion was enriched according to your comments.

We hope the answers we provided to your comments are satisfactory enough.

Thank you for your time and effort in reviewing our paper.

Sincerely,

A.S.

Round 2

Reviewer 1 Report

We thank the authors for responding well to the suggestions. The paper is worthy of considered publication after a minor revision. There is question that should be considered.

Line 20, please explain ‘CI’, and the mean is using confidence interval instead of abbreviation.

Line 623, ‘Canadian Journal of Public Health = Revue Canadienne de Sante Publique’.

Author Response

Respected reviewer,

Thank you for your comments on our paper.

Our answer to your comment in Line 20: We selected published research with different samples randomly from the same population (pregnant women) and computed a confidence interval for each sample to see how it may represent the true value of the population variable (vaccination acceptance). Since a statistically significant heterogeneity among studies was detected (I2= 99.29%%, p<0.001), an analysis was performed using the random effect model. The resulting datasets were all different; some intervals include the true population parameter and others do not. Confidence intervals measure the degree of uncertainty or certainty in the sampling method. In this Systematic Review, we took the probability limits, to be a 95% confidence level and our analysis revealed that the pooled proportion of vaccination uptake against COVID-19 in pregnant women was 0.53 (95% CI: 0.44 - 0.61).

In Line 623 it is corrected.

Yours

AS

Reviewer 2 Report

All comments were properly addressed.

Author Response

Respected reviewer,

Thank you for your kind remarks and we greatly appreciate the time you spent on our manuscript's review.

Yours,

AS